# Splicing factor SRSF1 is essential for homing of precursor spermatogonial stem cells in mice

Longjie Sun[1], Zheng Lv[1], Xuexue Chen[1], Rong Ye[2], Shuang Tian[1], Chaofan Wang[1], Xiaomei Xie[1], Lu Yan[1], Xiaohong Yao[1], Yujing Shao[1], Sheng Cui[3], Juan Chen[4]*, Jiali Liu[1]*

[1]State Key Laboratory of Animal Biotech Breeding, College of Biological Sciences, China Agricultural University, Beijing, China; [2]Key Laboratory of RNA Biology, Institute of Biophysics, Chinese Academy of Sciences, Beijing, China; [3]College of Veterinary Medicine, Yangzhou University, Jiangsu, China; [4]Key Laboratory of Precision Nutrition and Food Quality, Department of Nutrition and Health, China, Agricultural University, Beijing, China

*For correspondence:
chenjuan.09@163.com (JC);
liujiali@cau.edu.cn (JL)

Competing interest: The authors declare that no competing interests exist.

**Abstract** Spermatogonial stem cells (SSCs) are essential for continuous spermatogenesis and male fertility. The underlying mechanisms of alternative splicing (AS) in mouse SSCs are still largely unclear. We demonstrated that SRSF1 is essential for gene expression and splicing in mouse SSCs. Crosslinking immunoprecipitation and sequencing data revealed that spermatogonia-related genes (e.g. *Plzf*, *Id4*, *Setdb1*, *Stra8*, *Tial1/Tiar*, *Bcas2*, *Ddx5*, *Srsf10*, *Uhrf1*, and *Bud31*) were bound by SRSF1 in the mouse testes. Specific deletion of *Srsf1* in mouse germ cells impairs homing of precursor SSCs leading to male infertility. Whole-mount staining data showed the absence of germ cells in the testes of adult conditional knockout (cKO) mice, which indicates Sertoli cell-only syndrome in cKO mice. The expression of spermatogonia-related genes (e.g. *Gfra1*, *Pou5f1*, *Plzf*, *Dnd1*, *Stra8*, and *Taf4b*) was significantly reduced in the testes of cKO mice. Moreover, multiomics analysis suggests that SRSF1 may affect survival of spermatogonia by directly binding and regulating *Tial1/Tiar* expression through AS. In addition, immunoprecipitation mass spectrometry and co-immunoprecipitation data showed that SRSF1 interacts with RNA splicing-related proteins (e.g. SART1, RBM15, and SRSF10). Collectively, our data reveal the critical role of SRSF1 in spermatogonia survival, which may provide a framework to elucidate the molecular mechanisms of the posttranscriptional network underlying homing of precursor SSCs.

## eLife assessment

In this **valuable** study, the authors characterize the role of splicing factor SRSF1 during spermatogenesis with a conditional knockout of *Srsf1* in male germ cells. The phenotype and molecular role of SRSF1 in regulating alternative splicing in precursor spermatogonial stem cells in juvenile testes are **convincingly** supported. The paper also provides **convincing** evidence that the mRNA encoding Tial, a factor relevant to spermatogonial maintenance and male fertility, is alternatively spliced in testis and that this splicing is regulated by SRSF1. The work will be of interest to the fields of reproductive biology, stem cell biology, and alternative splicing.

## Introduction

Sertoli cell-only syndrome (SCOS), also known as del Castillo syndrome or germ cell aplasia, is one of the most common causes of severe non-obstructive azoospermia (NOA) (*Wang et al., 2023*). SCOS is the presence of only Sertoli cells in the testicular tubules of the testes, with no germ cells present (*Juul et al., 2014*; *Wang et al., 2023*). It is well known that abnormal self-renewal and differentiation of spermatogonial stem cells (SSCs) lead to SCOS (*Kanatsu-Shinohara and Shinohara, 2013*; *La and Hobbs, 2019*). In mice, gonocytes begin homing at 0–3 days postpartum (dpp) and then develop into SSCs at 4–6 dpp for continuous self-renewal and differentiation (*Lee and Shinohara, 2011*; *McLean et al., 2003*; *Tan and Wilkinson, 2020*). The mechanisms regulating homing of precursor SSCs are hence crucial for forming SSC pools and establishing niches (*Oatley and Brinster, 2012*). Spermatogonia migrate to form two distinct subtypes in mice. The first subtype develops into precursor SSCs that provide an SSC population for adult spermatogenesis, whereas the second subtype transitions directly to differentiated spermatogonia that contribute to the first round of spermatogenesis but do not self-renew (*Kluin and de Rooij, 1981*; *Law et al., 2019*). Therefore, homing of precursor SSCs to establish niches is essential for SSC self-renewal and differentiation.

Many transcription factors (e.g. FOXO1, PLZF, POU5F1, TAF4B, CHD4, BCL6B, BRACHYURY, ETV5, ID4, LHX1, POU3F1, DMRT1, NGN3, SOHLH1, SOHLH2, SOX3, and STAT3) promote SSC self-renewal and differentiation (*Cafe et al., 2021*; *Song and Wilkinson, 2014*). However, the molecular mechanisms of the posttranscriptional network underlying homing of precursor SSCs are not sufficiently clear. Previous studies have identified the key RNA-binding proteins DND1 and DDX5 in SSCs with a unique and dominant role in posttranscriptional regulation (*Legrand et al., 2019*; *Yamaji et al., 2017*). Surprisingly, recent studies have found that the RNA-binding proteins SRSF10, UHRF1, BUD31, and BCAS2 regulate alternative splicing (AS) in mouse spermatogonia (*Liu et al., 2022*; *Liu et al., 2017*; *Qin et al., 2023*; *Zhou et al., 2022*). It is well known that testes are rich in AS events (*Mazin et al., 2021*; *Venables, 2002*). Thus, understanding the mechanisms of AS in human reproduction can provide new insights into clinical diagnosis. However, the underlying mechanisms of how AS functions in homing of precursor SSCs are still largely unclear.

Serine/arginine-rich splicing factor 1 (SRSF1, previously SF2/ASF) is a widely studied and important splicing factor involved in cancer progression, heart development, and thymus development (*Du et al., 2021*; *Katsuyama et al., 2019*; *Katsuyama and Moulton, 2021*; *Liu et al., 2021*; *Lv et al., 2021*; *Qi et al., 2021*; *Xu et al., 2005*). Our previous work has shown that SRSF1 deficiency impairs primordial follicle formation during meiotic prophase I and leads to primary ovarian insufficiency (*Sun et al., 2023b*). However, the underlying mechanisms by which SRSF1 regulates pre-mRNA splicing in mouse SSCs remain unknown. A mouse model with *Srsf1* conditional deletion can effectively address this uncertainty. This study showed that specific deletion of *Srsf1* in mouse germ cells leads to NOA by impairing homing of mouse precursor SSCs. We further verified that SRSF1 directly binds and regulates *Tial1/Tiar* expression via AS, which may be critical for homing of mouse precursor SSCs.

## Results

### SRSF1 may have a vital role in posttranscriptional regulation in the testes

To investigate the role of SRSF1 in spermatogenesis, the dynamic localisation of SRSF1 in the testis was evaluated. Fascinatingly, the results of SRSF1 and γH2AX co-staining revealed that SRSF1 was expressed during spermatogenesis (*Figure 1A* and *Figure 1—figure supplement 1*). RT-qPCR and western blotting results showed that the expression of SRSF1 fluctuated during the developmental stages of the testes (*Figure 1B and C*). Concurrently, the results of SRSF1 and PLZF co-staining revealed that SRSF1 was highly expressed in the nuclei of spermatogonia (*Figure 1D*). To further explore the function of SRSF1 in regulating SSC self-renewal and differentiation, crosslinking immunoprecipitation and sequencing (CLIP-seq) was performed in adult mouse testes (*Sun et al., 2023a*). Gene Ontology (GO) enrichment analyses of the SRSF1 peak-containing genes revealed that spermatogenesis-related genes were regulated by SRSF1 (*Figure 2A* and *Supplementary file 1*). In combination with previous studies, we found that spermatogonia-related genes (e.g. *Plzf*, *Id4*, *Setdb1*, *Stra8*, *Tial1/Tiar*, *Bcas2*, *Ddx5*, *Srsf10*, *Uhrf1*, and *Bud31*) were bound by SRSF1. To provide in-depth insight into the binding of spermatogonia-associated genes, the SRSF1-binding peaks of the gene transcripts were shown by

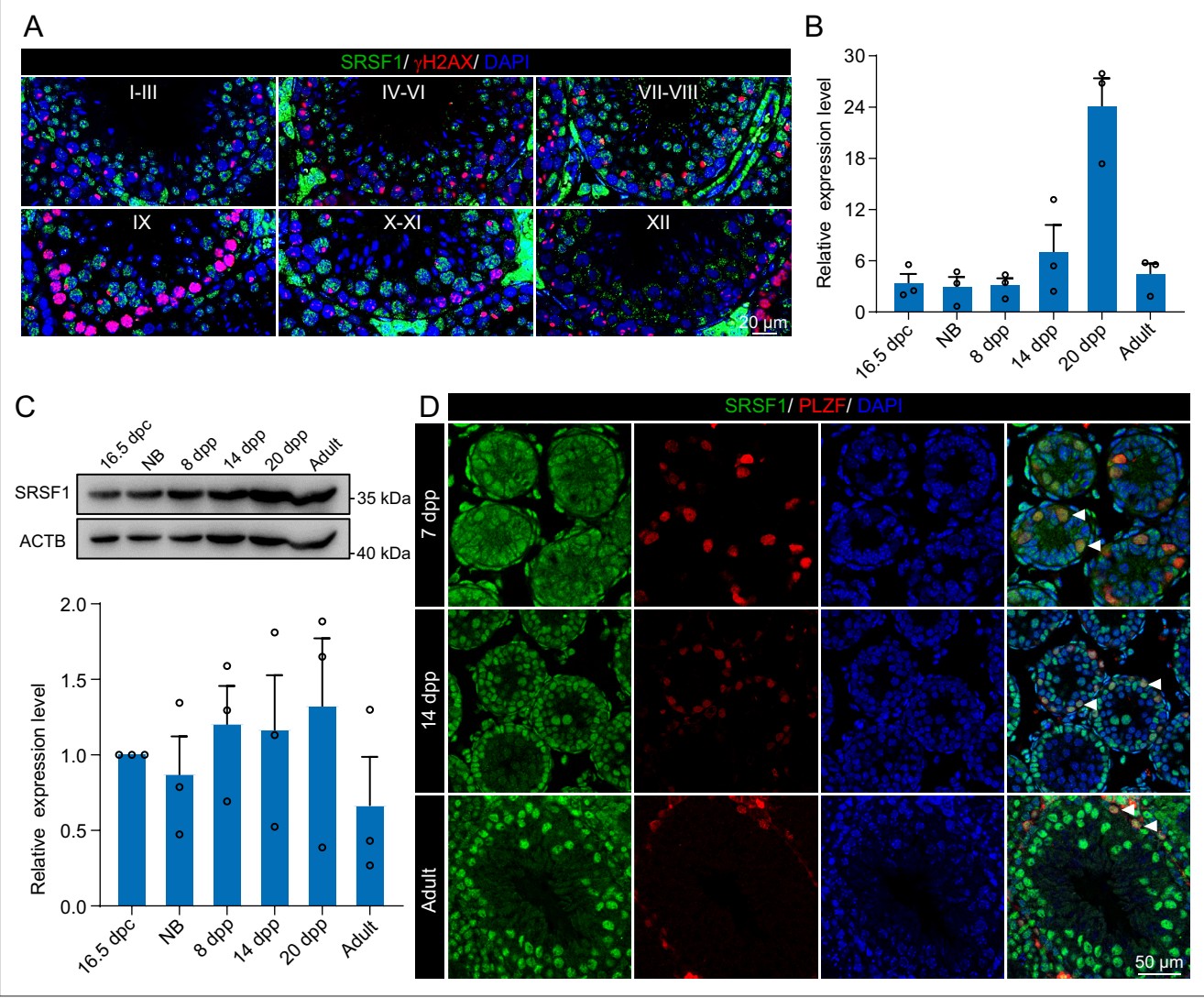

**Figure 1.** Expression and localisation of SRSF1 in the testis of mice at different developmental stages. (**A**) Dynamic localisation of SRSF1 during spermatogenesis. Co-immunostaining was performed using SRSF1 and γH2AX antibodies in adult mouse testes. DNA was stained with DAPI. Scale bar, 20 μm. (**B**) Expression of *Srsf1* in testes at different stages of development. The RT-qPCR data were normalised to *Gapdh*. N=3. (**C**) Western blotting of SRSF1 expression in testes at different stages of development. ACTB served as a loading control. The value in 16.5 days post-coitus (dpc) testes were set as 1.0, and the relative values of testes in other developmental periods are indicated. N=3. (**D**) Localisation and expression of SRSF1 in spermatogonia. Co-immunostaining was performed using PLZF and SRSF1 antibodies in 7 days postpartum (dpp), 14 dpp, and adult mouse testes. DNA was stained with DAPI. Arrowheads, spermatogonia. Scale bar, 50 μm.

The online version of this article includes the following source data and figure supplement(s) for figure 1:

**Source data 1.** Western blotting of SRSF1 expression in testes at different stages of development.

**Figure supplement 1.** Dynamic localisation of SRSF1 during spermatogenesis.

using Integrative Genomics Viewer (IGV) (*Figure 2B*). The co-staining results showed localisation and expression of the spermatogonia-related proteins in mouse testes (*Figure 2C*). Together, these results suggested that SRSF1 may have a vital role in posttranscriptional regulation in the testes, particularly during spermatogonial development.

## SRSF1 deficiency leads to SCOS

To define the specific involvement of SRSF1 in spermatogonia, we studied the physiological roles of SRSF1 in vivo using a mouse model. Considering that global *Srsf1* knockout is lethal in mice (*Xu et al., 2005*), we used a conditional allele of *Srsf1* (*Srsf1*^Fl^) in which exons 2, 3, and 4 of *Srsf1* are flanked by

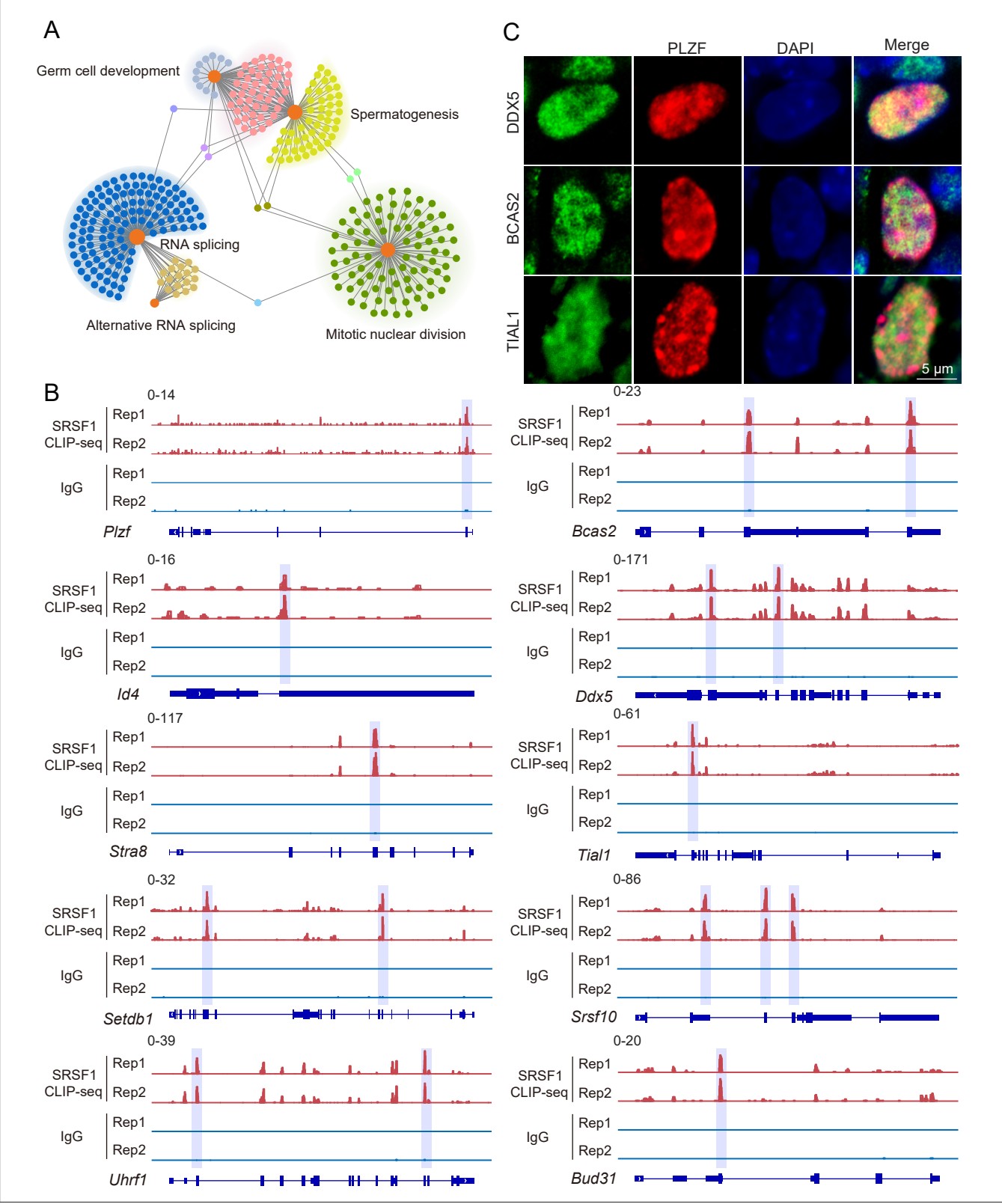

**Figure 2.** SRSF1-binding genes have an essential role in spermatogonia. (**A**) Network showing Gene Ontology (GO) enrichment analyses of SRSF1-binding genes. (**B**) Representative genome browser views of spermatogonia-related gene transcripts bound by SRSF1. Higher peaks are marked by a lavender area. (**C**) Localisation of the spermatogonia-related proteins in mouse testes. Scale bar, 5 µm.

two *loxP* sites (*Figure 3A*). By crossing *Srsf1*^Fl and *Vasa*-Cre mice, we obtained *Vasa*-Cre *Srsf1*^Fl/del mice with *Srsf1* deletion in germ cells (*Figure 3A and B*). We verified the absence of the SRSF1 protein in germ cells by co-immunofluorescence analyses with SRSF1 and PLZF antibodies (*Figure 3C*). Subsequently, the breeding experiment indicated that conditional knockout (cKO) mice had a standard mating capacity but that the absence of *Srsf1* led to complete infertility in males (*Figure 3D*). Histological examination of cKO epididymides revealed that sperm could not be found in the cauda epididymis (*Figure 3E*). Considering the limitations of sectioning, the cauda epididymal sperm count further validated this conclusion (*Figure 3F*). It was clear that spermatogenesis in the testes was severely impaired. Therefore, we focused our attention on the testes. The adult cKO mice were normal in size (*Figure 3G*). However, the sizes of cKO mouse testes were significantly reduced (*Figure 3H*). Histological examination of cKO testis sections showed that no germ cells could be visualised, and only a large number of Sertoli cells were observed in the testes of cKO mice (*Figure 3I*). Together, these results demonstrated that SRSF1 is critical for spermatogenesis and that its absence leads to SCOS.

## Loss of SRSF1 impairs spermatogonia survival

To further confirm the absence of germ cells in the testes of cKO mice, PLZF and γH2AX co-staining was performed in adult mouse testes. These data suggested that SRSF1 deficiency impaired germ cell survival (*Figure 4A*). The results of VASA and TRA98 co-staining further confirmed this phenotype (*Figure 4B*). Considering the limitations of sectioning, we used whole-mount immunostaining to perform a comprehensive analysis and found that germ cells were indeed absent in the testes of adult cKO mice (*Figure 4C*). To dynamically analyse the loss of germ cells, we collected testes from 5 dpp, 7 dpp, and 14 dpp mice. Morphological results showed that the testes of 7 dpp and 14 dpp cKO mice were much smaller than those of Ctrl mice (*Figure 5A*). To determine the presence of germ cells in cKO testes, VASA staining was performed in 5 dpp, 7 dpp, and 14 dpp Ctrl and cKO testes. The results showed that germ cells were still present in cKO mice but were significantly reduced in 7 dpp and 14 dpp cKO testes (*Figure 5B*). The quantifications of germ cells per tubule showed a significant reduction in the number of 7 dpp and 14dpp cKO testes, especially 14 dpp cKO testes (*Figure 5B*). In addition, TUNEL results showed that apoptosis significantly increased in cKO testes (*Figure 5C*). These data suggested that the absence of SRSF1 causes apoptosis in a large number of spermatogonia that are unable to survive.

## Loss of SRSF1 impairs homing of precursor SSCs

To further investigate the reason for the failure of spermatogonia to survive, we observed homing of precursor SSCs in the testes of mice at 5 dpp. Interestingly, the results of VASA and SOX9 co-staining showed that partial germ cells could not complete homing in 5 dpp cKO testes (*Figure 6A and B*). In mice, starting at 3 dpp, cytoplasmic FOXO1 in some gonocytes begins to enter the nucleus (*Goertz et al., 2011*). These cells further develop into prospermatogonia, which are expected to develop into SSCs (*Goertz et al., 2011*). Thus, immunohistochemical staining for FOXO1 was performed on 5 dpp mouse testis sections (*Figure 6C*). Further, germ cell statistics of FOXO1 expression in the nucleus showed a reduced number of prospermatogonia in cKO mice (*Figure 6D*). And germ cells in which FOXO1 is expressed in the nucleus similarly undergo abnormal homing (*Figure 6E*). Thus, all the above data indicated that SRSF1 has an essential role in the homing of precursor SSCs.

## SRSF1 is essential for gene expression in spermatogonia

To investigate the molecular mechanisms of SRSF1 in spermatogonia, we isolated mRNA from 5 dpp cKO and Ctrl mouse testes and performed RNA-seq (*Figure 7—figure supplement 1*). RNA-seq and RT-qPCR results showed a significant reduction in the expression of *Srsf1* in 5 dpp cKO mouse testes (*Figure 7A*). Western blotting results showed that SRSF1 expression was significantly reduced in the testes of cKO mice at 5 dpp (*Figure 7B*). Hence, for Ctrl and cKO samples, the confidence level of the RNA-seq data was high. The volcano map and cluster heatmap showed 715 downregulated and 258 upregulated genes identified by RNA-seq data in 5 dpp cKO mouse testes (*Figure 7C and D* and *Supplementary file 2*). These gene GO enrichment analyses indicated abnormal germ cell development and cell cycle arrest in 5 dpp cKO mouse testes (*Figure 7E*). Surprisingly, the heatmap showed that spermatogonia-associated gene (e.g. *Gfra1*, *Pou5f1*, *Plzf*, *Nanos3*, *Dnd1*, *Stra8*, and *Taf4b*) expression was significantly reduced in the testes of cKO mice at 5 dpp (*Figure 7F*). Simultaneously,

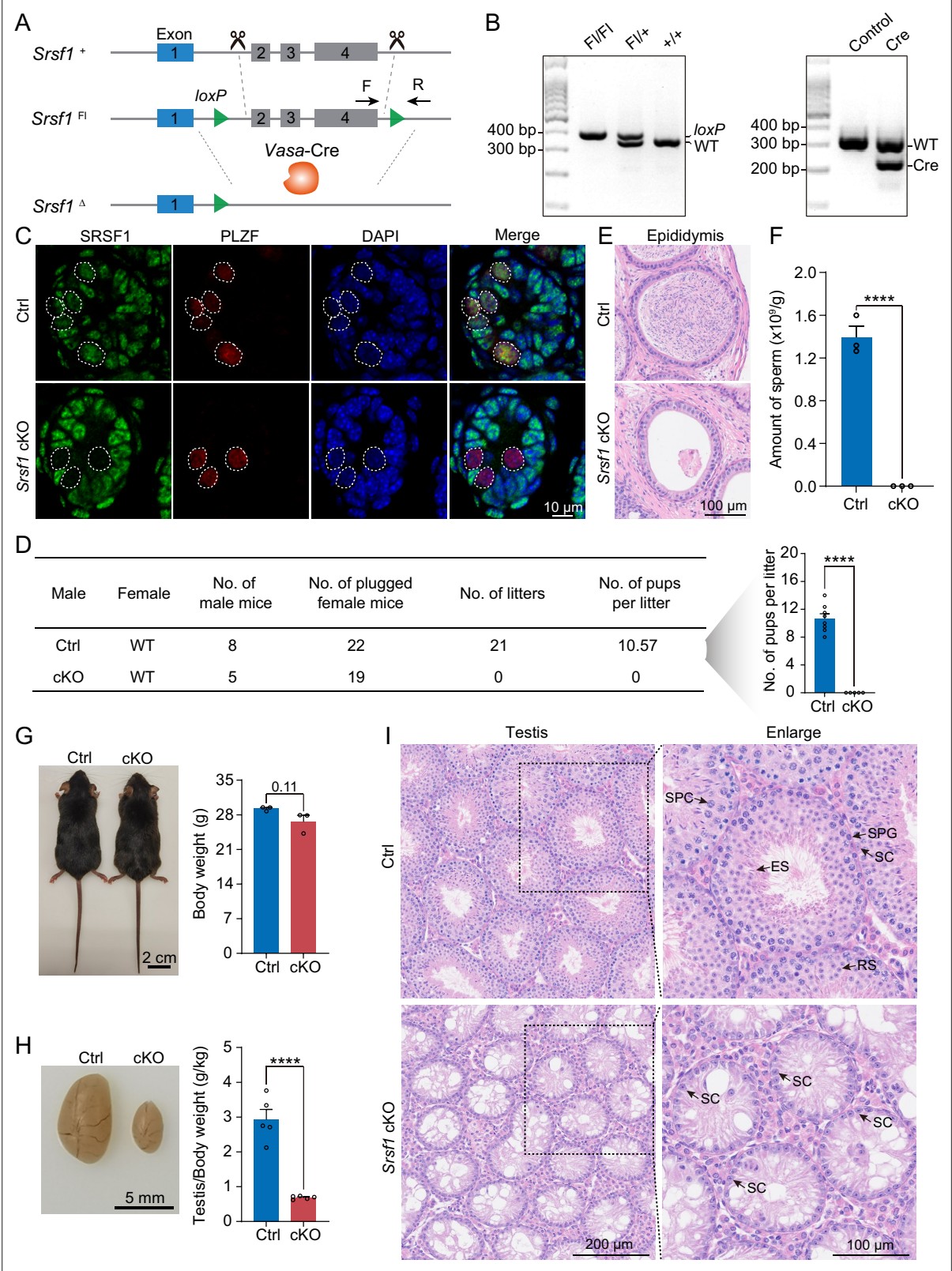

**Figure 3.** SRSF1 plays critical roles in spermatogenesis and male fertility. (**A**) *Vasa*-Cre mice were crossed with *Srsf1*^Fl/Fl mice to generate *Srsf1* conditional knockout (cKO) mice. Cre-mediated deletion removed exons 2, 3, and 4 of *Srsf1* and generated a null protein allele. (**B**) Genotyping PCR was performed using *Vasa*-Cre and *Srsf1* primers. (**C**) Co-immunostaining of SRSF1 and PLZF in 7 days postpartum (dpp) control (Ctrl) and cKO testis. DNA was stained with DAPI. Scale bar, 10 μm. (**D**) Fertility test results showed a male infertility phenotype in the cKO mice (N=5) compared to the Ctrl mice (N=8). The

*Figure 3 continued on next page*

Figure 3 continued

number of pups per litter was determined in the cKO (N=5) and Ctrl (N=8) mice. (**E**) Haematoxylin-eosin-stained epididymis sections from adult Ctrl and cKO mice were obtained. Scale bar, 100 μm. (**F**) Cauda epididymal sperm counting was performed. N=3. (**G**) Normal body weight in cKO mice. The body sizes and weights of adult Ctrl and cKO mice are shown as the mean ± SEM. N=3. (**H**) Testis atrophy in adult cKO mice. Testis sizes and weights/ body of adult Ctrl and cKO mice are shown as the mean ± SEM. N=5. (**I**) Haematoxylin-eosin-stained testis sections from adult Ctrl and cKO mice were obtained. Scale bar, left panel: 200 μm, right panel: 100 μm. SC, Sertoli cell; SPG, spermatogonium; SPC, spermatocyte; RS, round spermatid; ES, elongated spermatid. Unpaired Student's *t*-test determined significance; exact p value p≥0.05, ****p<0.0001. The points and error bars represent the mean ± SEM.

The online version of this article includes the following source data for figure 3:

**Source data 1.** Genotyping PCR was performed using *Vasa*-Cre and *Srsf1* primers.

the tracks of differentially expressed genes are demonstrated by using IGV (*Figure 7G*). Next, we validated the abnormal expression of spermatogonia-associated genes (downregulated: *Gfra1*, *Pou5f1*, *Plzf*, *Dnd1*, *Stra8*, and *Taf4b*; unchanged: *Nanos3*) by RT-qPCR (*Figure 7H*). Together, these data indicated that germ cell-specific deletion of *Srsf1* impairs the expression of spermatogonia-associated genes.

## SRSF1 directly binds and regulates the expression and AS of *Tial1/Tiar*

Multiomics analyses were carried out in a subsequent study to identify the molecular mechanisms by which SRSF1 regulates spermatogonia survival. Considering that the CLIP-seq data were obtained from adult mouse testis, the set of genes bound by CLIP-seq was restricted to those expressed only in the 5 dpp mouse testis RNA-seq data. we found that 3543 of the 4824 genes bound by SRSF1 were expressed in testes at 5 dpp. Venn diagram data revealed that 9 out of 715 downregulated genes were bound by SRSF1 and underwent abnormal AS (*Figure 8A*). And 1 out of 258 upregulated genes was bound by SRSF1 and underwent abnormal AS (*Figure 8A*). Interestingly, we found that 39 unchanged genes were bound by SRSF1 and underwent abnormal AS (*Figure 8A*). The AS genes were subsequently investigated in 5 dpp cKO mouse testes using transcriptomic analyses. RNA-seq analyses showed that 162 AS events were significantly affected (false discovery rate [FDR] <0.05) in cKO mouse testes (*Figure 8B and C* and *Supplementary file 3*). Most of the 133 affected AS events (162) were classified as skipped exons (SEs), with 10 AS events categorised as retained introns (RIs), 13 as mutually exclusive exons (MXEs), 4 as alternative 5' splice sites (A5SSs), and 2 as alternative 3' splice sites (A3SSs) (*Figure 8C*). Additionally, the overall analysis of aberrant AS events showed that SRSF1 effectively inhibits the occurrence of SE and MXE events and promotes the occurrence of RI events (*Figure 8C*). Then, GO enrichment analyses of AS genes revealed that four genes concerning germ cell development were altered in AS forms (*Figure 8D*). It has been shown that *Tial1/Tiar* affects the survival of primordial germ cells (*Beck et al., 1998*). Moreover, *Tial1/Tiar* is 1 of 39 unchanged genes that are bound by SRSF1 and undergo abnormal AS. Thus, multiomics analyses suggested that *Tial1/Tiar* were posttranscriptionally regulated by SRSF1. Next, we investigated the mechanism by which SRSF1 regulates the AS of *Tial1/Tiar*, RT-PCR results showed that the pre-mRNA of *Tial1/Tiar* in 5 dpp cKO mouse testes exhibited abnormal AS (*Figure 8E*). We then visualised the different types of AS based on RNA-seq data by using IGV (*Figure 8F* and *Figure 8—figure supplement 1*). The results of RNA immunoprecipitation (RIP)-qPCR showed that SRSF1 could bind to the pre-mRNA of *Tial1/Tiar* (*Figure 8G*). Interestingly, RNA-seq analyses showed that the fragments per kilobase million (FPKM) of *Tial1/Tiar* was unchanged in 5 dpp cKO mouse testes (*Figure 8H*). RT-qPCR results showed that *Tial1/Tiar* transcript levels were not inhibited (*Figure 8I*). However, western blotting showed that expression levels of TIAL1/TIAR isoform X2 were significantly suppressed (*Figure 8J* and *Figure 8—figure supplement 1*). In summary, the data indicate that SRSF1 is required for TIAL1/TIAR expression and splicing in spermatogonia survival.

## SRSF1 recruits AS-related proteins to modulate AS in testes

To identify the interacting proteins for which SRSF1 exerts its AS role, we performed MS analyses of IP samples from 5 dpp mouse testis extracts. The silver-stained gel of SRSF1 and normal IgG showed several SRSF1-interacting proteins from 5 dpp mouse testis extracts (*Figure 9A*). The IP results indicated that SRSF1 was able to effectively IP the testis extracts of 5 dpp mice (*Figure 9B*). Immunoprecipitation mass spectrometry (IP-MS) data demonstrated the efficient enrichment of SRSF1 (*Figure 9C*

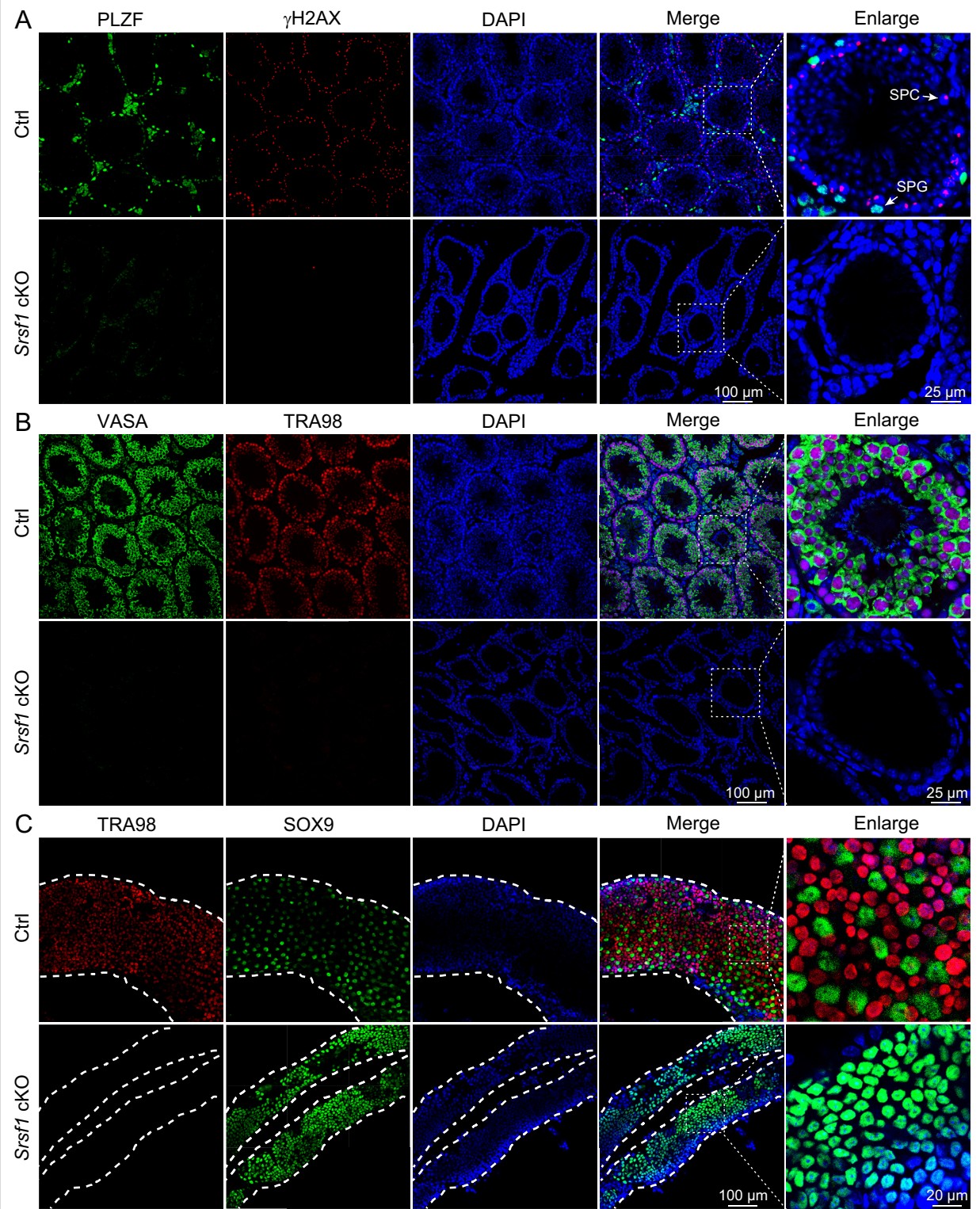

**Figure 4.** Loss of germ cells in adult conditional knockout (cKO) mouse testes. (**A**) Co-immunostaining of PLZF and γH2AX in adult control (Ctrl) and cKO testis. DNA was stained with DAPI. Scale bar, right panel: 25 μm, other panels: 100 μm. (**B**) Co-immunostaining of VASA and TRA98 in adult Ctrl and cKO testis. DNA was stained with DAPI. Scale bar, right panel: 25 μm, other panels: 100 μm. (**C**) Whole-mount co-immunostaining of TRA98 and SOX9 in adult Ctrl and cKO testis. DNA was stained with DAPI. White dashed lines, boundary of the tubule. Scale bar, right panel: 20 μm, other panels: 100 μm.

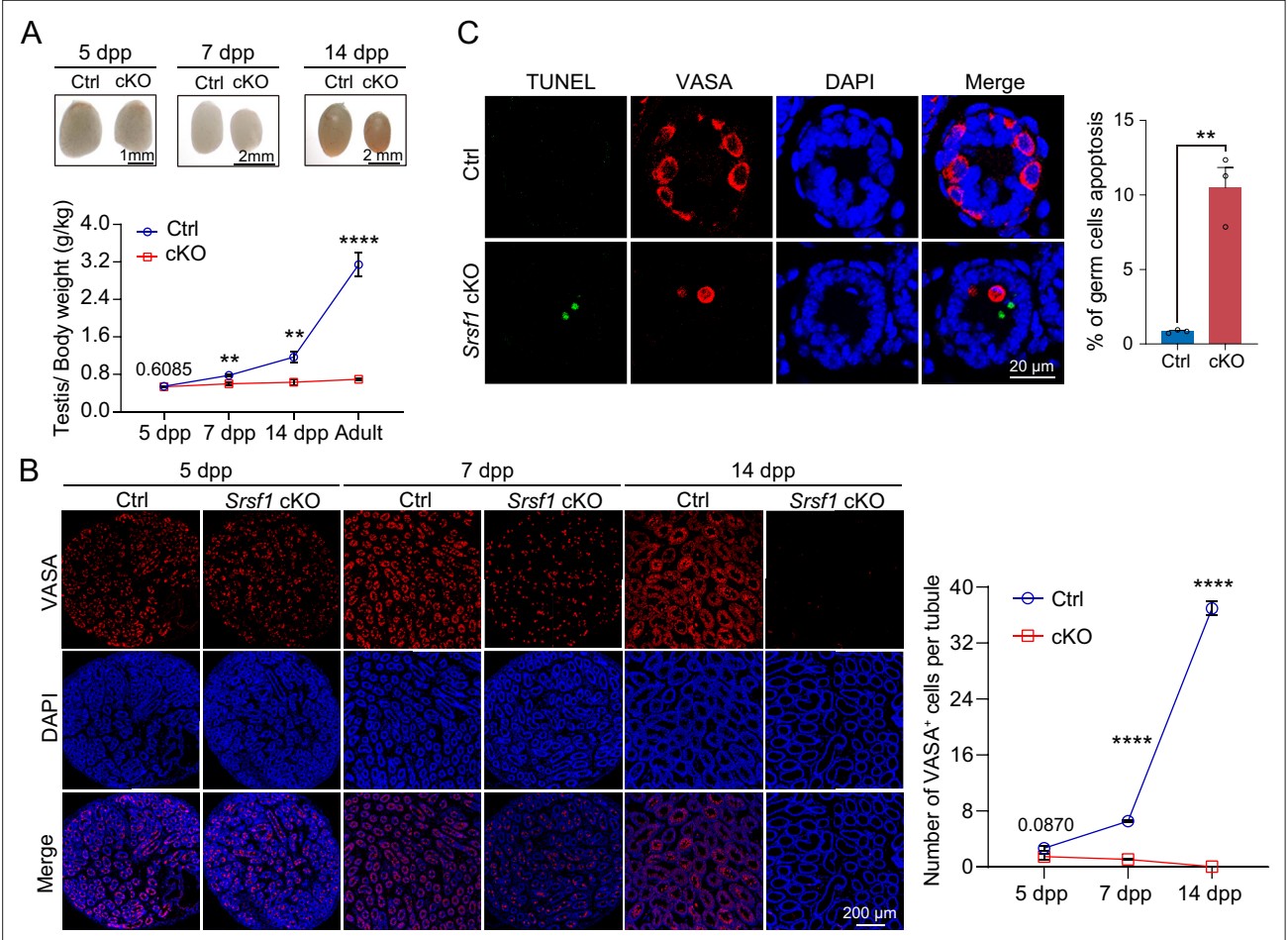

**Figure 5.** SRSF1 is required for spermatogonia survival. (**A**) Testis sizes of 5 days postpartum (dpp), 7 dpp, and 14 dpp control (Ctrl) and conditional knockout (cKO) mice are shown. The testis/body weight ratios (g/kg) of 5 dpp, 7 dpp, 14 dpp, and adult Ctrl and cKO mice are shown as the mean ± SEM. N=4. (**B**) Immunostaining of VASA in 5 dpp, 7 dpp, and 14 dpp Ctrl and cKO testis. DNA was stained with DAPI. Scale bar, 200 μm. Number of VASA-positive cells per tubule is the mean ± SEM. N=3. (**C**) TUNEL apoptosis assay was performed on sections from 7 dpp Ctrl and cKO testis. DNA was stained with DAPI. Scale bar, 20 μm. Percentage of germ cells apoptosis is the mean ± SEM. N=3. Unpaired Student's *t*-test determined significance; exact p value p≥0.05, **p<0.01, ****p<0.0001. The points and error bars represent the mean ± SEM.

and *Supplementary file 4*). These data showed that the two samples were highly reproducible, especially for SRSF1 (*Figure 9D*). Then, GO enrichment analyses of the IP proteins revealed that AS-related proteins could interact with SRSF1 (*Figure 9E*). A circular heatmap showed that SRSF1 could interact with AS-related proteins (e.g. SRSF10, SART1, RBM15, SRRM2, SF3B6, and SF3A2) (*Figure 9F*). The co-immunoprecipitation (Co-IP) results indicated that FLAG-SRSF1 interacted with HA-SART1, HA-RBM15, and HA-SRSF10 in 293T cells (*Figure 9G*). In addition, the fluorescence results showed complete co-localisation of mCherry-SRSF1 with eGFP-SART1, eGFP-RBM15, and eGFP-SRSF10 in 293T cells (*Figure 9H*). Co-IP suggested that the RRM1 domain of SRSF1 interacted with HA-SART1, HA-RBM15, and HA-SRSF10 in 293T cells (*Figure 9I*). Determining the complex structures of these interactions is valuable, in which molecular docking has played an important role (*Yan et al., 2017*). HDOCK is a novel web server of our hybrid docking algorithm of template-based modelling and free docking (*Yan et al., 2017*). The HDOCK analysis results depicted the RRM1 domain of SRSF1 with SRSF10, SART1, and RBM15 docking based on a hybrid strategy (*Figure 9J*). Together, the above data show that SRSF1 may interact with SRSF10, SART1, and RBM15 to regulate AS in 5 dpp mouse testes.

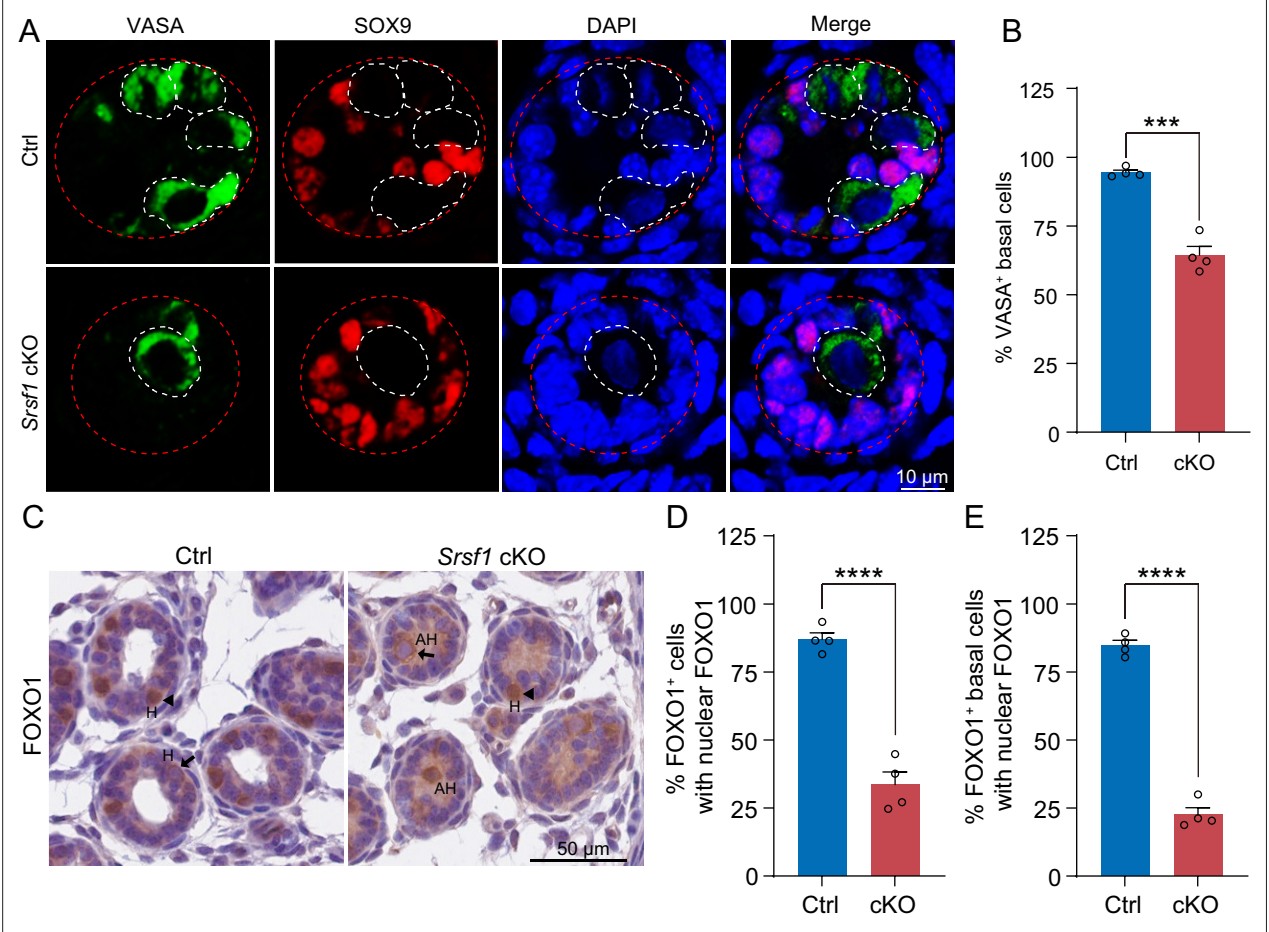

**Figure 6.** SRSF1 is required for homing of precursor spermatogonial stem cells (SSCs). (**A**) Co-immunostaining of VASA and SOX9 in 5 days postpartum (dpp) control (Ctrl) and conditional knockout (cKO) testis. DNA was stained with DAPI. Scale bar, 10 µm. Red dashed circles, tubule. White dashed circles, germ cell. (**B**) The percentage of VASA-positive basal cells is shown as the mean ± SEM. N=4. (**C**) Immunohistochemical staining of FOXO1 in 5 dpp Ctrl and cKO testis. The nuclei were stained with haematoxylin. Scale bar, 10 µm. Arrowheads, FOXO1 in the nucleus. Arrows, FOXO1 in the cytoplasm. H, homing. AH, abnormal homing. (**D**) The percentage of FOXO1-positive cells with nuclear FOXO1 is shown as the mean ± SEM. 913 Ctrl germ cells and 592 cKO germ cells were obtained from four animals. (**E**) The percentage of FOXO1-positive basal cells with nuclear FOXO1 is shown as the mean ± SEM. 913 Ctrl germ cells and 592 cKO germ cells were obtained from four animals. Unpaired Student's $t$-test determined significance; ***$p < 0.001$, ****$p < 0.0001$. The points and error bars represent the mean ± SEM.

## Discussion

### Failure of spermatogonia survival led to SCOS

Disturbed spermatogenesis can cause SCOS and ultimately male sterility (*Jiao et al., 2021*). In recent years, it has been reported that many spermatogonia-related gene deletions have disrupted SSC self-renewal and differentiation in patient and mouse models (*La and Hobbs, 2019*; *Tan and Wilkinson, 2020*; *Wang et al., 2021*). SCOS was observed in *Ddx5*, *Tial1/Tiar*, *Uhrf1*, *Pramef12*, *Dot1l*, and *Rad51* deletion mouse models (*Beck et al., 1998*; *Legrand et al., 2019*; *Lin et al., 2022*; *Qin et al., 2022*; *Wang et al., 2019*; *Zhou et al., 2022*). Mouse models are still of great significance and reference for human SCOS studies, and they will provide a better understanding of how SCOS occurs and develops over time. Interestingly, our mouse model had SCOS (*Figure 3D–I* and *Figure 4*). The absence of germ cells represents classical SCOS in adult mouse testes (*Figure 4*; *Wang et al., 2023*). In addition, we found abnormal expression of spermatogonia-related genes (e.g. *Gfra1*, *Pou5f1*, *Plzf*, *Dnd1*, *Stra8*, and *Taf4b*) in cKO mouse testes (*Figure 7F–H*). These differentially expressed genes regulate SSC self-renewal and differentiation in mouse testes (*Kanatsu-Shinohara and Shinohara, 2013*; *La and Hobbs, 2019*; *Tan and Wilkinson, 2020*). Thus, this provided an opportunity for us to better study the

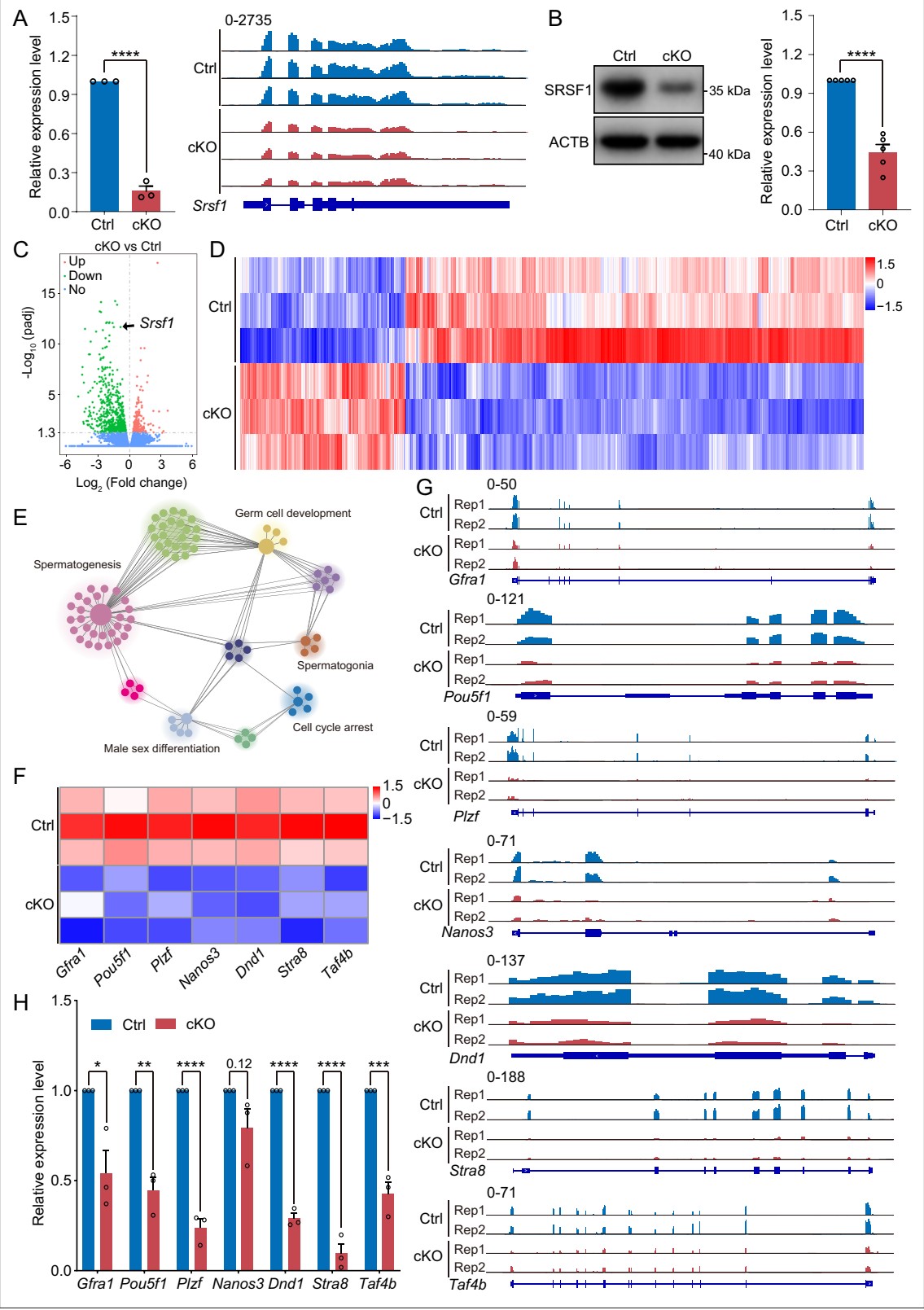

**Figure 7.** SRSF1 regulates the expression of spermatogonia-related genes. (**A**) Expression of *Srsf1* in 5 days postpartum (dpp) control (Ctrl) and conditional knockout (cKO) mouse testis. The RT-qPCR data were normalised to *Gapdh*. N=5. The expression of *Srsf1* is shown as reading coverage in 5 dpp mouse testis. (**B**) Western blotting of SRSF1 expression in 5 dpp mouse testis. ACTB served as a loading control. The value in Ctrl testes was set as 1.0, and the relative values in cKO testis are indicated. N=5. (**C**) Volcano map displaying the distribution of differentially expressed genes from RNA-

*Figure 7 continued on next page*

*Figure 7 continued*

seq data. The abscissa in the figure represents the gene fold change in 5 dpp cKO and Ctrl mouse testis. |log2FoldChange|≥0. The ordinate indicates the significance of gene expression differences between 5 dpp cKO and Ctrl mouse testis. padj ≤0.05. Upregulated genes are shown as red dots, and downregulated genes are shown as green dots. (**D**) Cluster heatmap of differentially expressed genes. The ordinate is the genotype, and the abscissa is the normalised FPKM (fragments per kilobase million) value of the differentially expressed gene. Red indicates a higher expression level, while blue indicates a lower expression level. (**E**) Network showing Gene Ontology (GO) enrichment analyses of differentially expressed genes. (**F**) Heatmap of spermatogonia-related gene expression. (**G**) The expression of spermatogonia-related genes is shown as read coverage. (**H**) The expression of spermatogonia-related genes in 5 dpp cKO and Ctrl mouse testis. The RT-qPCR data were normalised to *Gapdh*. The value in the Ctrl group was set as 1.0, and the relative value in the cKO group is indicated. N=3. Unpaired Student's *t*-test determined significance; exact p value p≥0.05, *p<0.05, **p<0.01, ***p<0.001, ****p<0.0001. The points and error bars represent the mean ± SEM.

The online version of this article includes the following source data and figure supplement(s) for figure 7:

**Source data 1.** Western blotting of SRSF1 expression in 5 days postpartum (dpp) mouse testes.

**Figure supplement 1.** The libraries of RNA-seq are of good quality.

underlying molecular mechanisms. These data indicate that SRSF1 deficiency impairs spermatogonia survival, leading to SCOS in male mice.

## The formation of SSC pools and the establishment of niches are essential for spermatogenesis

The earliest event in the development of the SSC population is the migration of prospermatogonia from the centre of seminiferous cords where they have resided since sex determination of the embryonic gonad to the basement membrane (*Oatley and Brinster, 2012*). In mice, this process is also known as homing of precursor SSCs, which occurs in the initial 3 dpp and then develops into SSCs at 4–6 dpp for continuous self-renewal and differentiation (*Lee and Shinohara, 2011*; *McLean et al., 2003*; *Oatley and Brinster, 2012*; *Tan and Wilkinson, 2020*). Therefore, homing analysis of precursor SSCs was performed in 5 dpp cKO mouse testes. Interestingly, the VASA and SOX9 co-staining results demonstrated that partial germ cells could not complete homing in 5 dpp cKO testes (*Figure 6A and B*). Further, immunohistochemical staining for FOXO1 and statistical results indicated that germ cells in which FOXO1 is expressed in the nucleus similarly undergo abnormal homing (*Figure 6C–E*). Germ cells that do not migrate to the basement membrane are unable to form SSC pools and establish niches (*McLean et al., 2003*). These SSCs that lose their ecological niche will cease to exist. In our data, TUNEL results showed that apoptosis significantly increased in 7 dpp cKO mouse testes. At once, the quantifications of germ cells per tubule showed a significant reduction in 7 dpp and 14 dpp cKO testes, especially 14 dpp cKO testes (*Figure 5B*). In conclusion, SRSF1 is crucial for the formation of SSC pools and the establishment of niches through homing of precursor SSCs.

## Abnormal AS impaired the survival of spermatogonia

AS is commonly found in mammals, especially in the brain and testes (*Mazin et al., 2021*; *Merkin et al., 2012*; *Wang et al., 2008*). AS plays essential roles in the posttranscriptional regulation of gene expression during many developmental processes, such as SSC self-renewal and differentiation (*Chen et al., 2018*; *Song et al., 2020*). Recently, BUD31-mediated AS of *Cdk2* was shown to be required for SSC self-renewal and differentiation (*Qin et al., 2023*). *Srsf10* depletion disturbed the AS of genes, including *Nasp*, *Bclaf1*, *Rif1*, *Dazl*, *Kit*, *Ret*, and *Sycp1* (*Liu et al., 2022*). UHRF1 interacts with snRNAs and regulates AS of *Tle3* in mouse SSCs (*Zhou et al., 2022*). Mettl3-mediated m6A regulates AS of *Sohlh1* and *Dazl* (*Xu et al., 2017*). We found that SRSF1 acts as an alternative RNA splicing regulator and directly interacts with *Tial1/Tiar* transcripts to regulate splicing events in spermatogonia (*Figure 8E–G*). Additionally, expression levels of TIAL1/TIAR isoform X2 were significantly suppressed (*Figure 8J*). Interestingly, *Tial1/Tiar* transcript levels were not inhibited (*Figure 8H and I*). These results suggested that SRSF1 explicitly regulates the expression of *Tial1/Tiar* via AS. Studies have reported that TIAL1/TIAR is essential for primordial germ cell development in mouse testes (*Beck et al., 1998*). *Tial1/Tiar* deletion impairs germ cell survival leading to SCOS, consistent with our phenotype (*Figures 3E–I and 4A–C*; *Beck et al., 1998*). Taken together, SRSF1 may affect germ cell survival by directly binding and regulating *Tial1/Tiar* expression through AS.

We found that SRSF1 could interact with AS-related proteins (e.g. SRSF10, SART1, RBM15, SRRM2, SF3B6, and SF3A2) (*Figure 9F*). A recent study reported that SRSF10 deficiency impaired

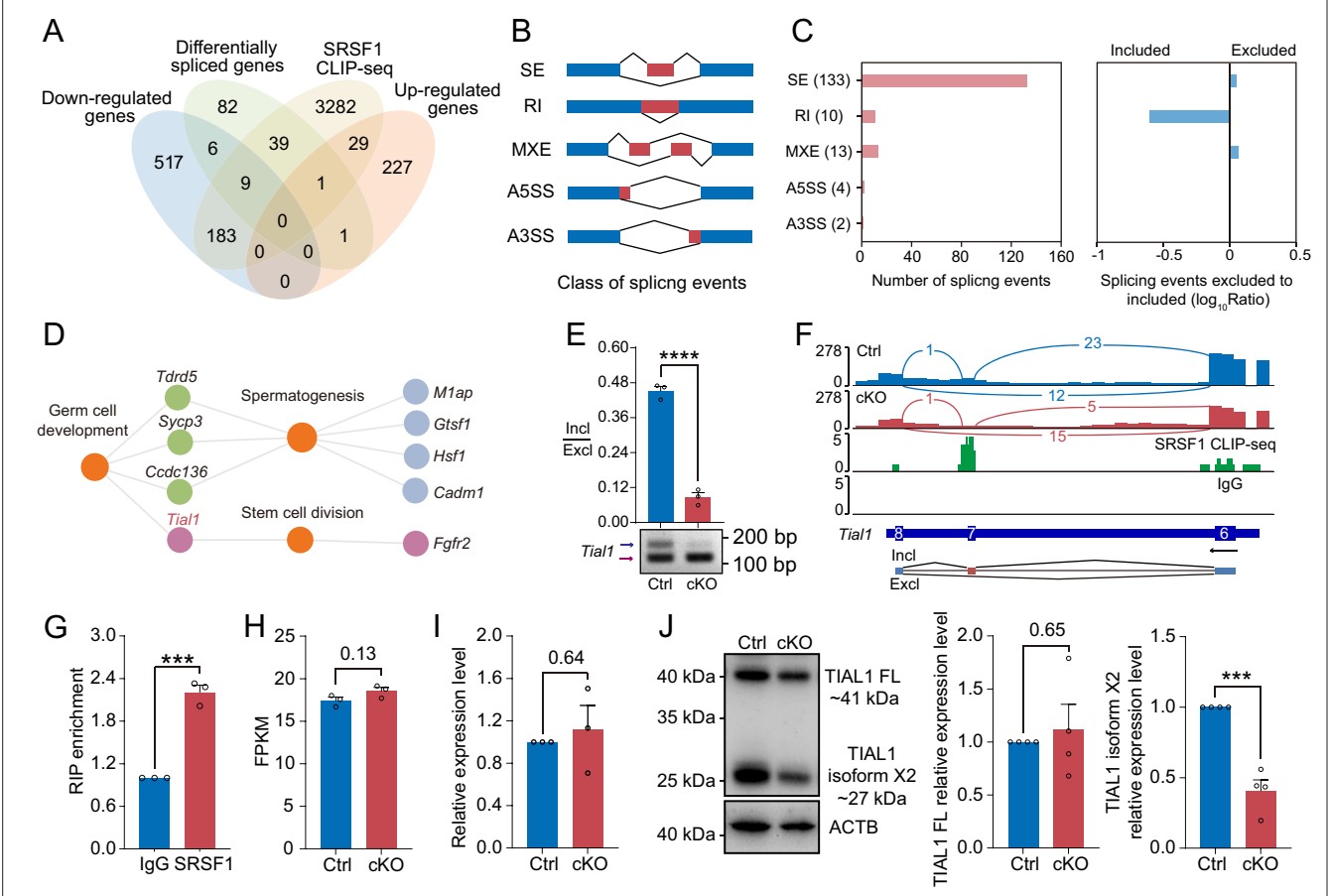

**Figure 8.** SRSF1 directly binds and regulates the expression and alternative splicing (AS) of *Tial1/Tiar*. (**A**) Venn diagram showing the correlation among downregulated, upregulated, alternatively spliced, and SRSF1-binding genes. (**B**) Schematic diagram showing the classes of splicing events. (**C**) Splicing events were analysed by number, exclusion, and inclusion. (**D**) Network showing Gene Ontology (GO) enrichment analyses of AS genes. (**E**) The AS of *Tial1/Tiar* in 5 days postpartum (dpp) Ctrl and conditional knockout (cKO) mouse testes was analysed by RT-PCR. N=3. The ratio of inclusion (Incl) to exclusion (Excl) is shown accordingly. (**F**) Analyses of *Tial1/Tiar* expression and exon-exon junctions were performed. (**G**) SRSF1 directly binds the pre-mRNA of *Tial1/Tiar* by RNA immunoprecipitation (RIP)-qPCR in 5 dpp mouse testes. N=3. (**H**) The fragments per kilobase million (FPKM) of *Tial1/Tiar* in 5 dpp Ctrl and cKO mouse testes. (**I**) The expression of *Tial1/Tiar* in 5 dpp Ctrl and cKO mouse testes. The RT-qPCR data were normalised to *Gapdh*. The value in the Ctrl group was set as 1.0, and the relative value in the cKO group is indicated. N=3. (**J**) Western blotting of TIAL1/TIAR expression in 5 dpp mouse testes. ACTB served as a loading control. The FL/isoform X2 value of TIAL1/TIAR in the Ctrl group was set as 1.0, and the relative value in the cKO group is indicated. N=3. FL, full length. Unpaired Student's *t*-test determined significance; exact p value p≥0.05, ***p<0.001, ****p<0.0001. The points and error bars represent the mean ± SEM.

The online version of this article includes the following source data and figure supplement(s) for figure 8:

**Source data 1.** The alternative splicing (AS) of *Tial1/Tiar* in 5 days postpartum (dpp) Ctrl and conditional knockout (cKO) mouse testes was analysed by RT-PCR.

**Source data 2.** Western blotting of TIAL1/TIAR expression in 5 days postpartum (dpp) mouse testes.

**Figure supplement 1.** Transcription and translation information for *Tial1/Tiar*.

spermatogonia differentiation but did not affect homing of precursor SSCs (*Liu et al., 2022*). However, our data showed that SRSF1 is essential for homing of mouse precursor SSCs. Therefore, this suggests that SRSF1 has a specific function in the homing of precursor SSCs independent of SRSF10.

## SRSF1-mediated posttranscriptional regulation during homing of precursor SSCs provides new insights into the treatment of human reproductive diseases

Aberrant homing of precursor SSCs often lead to gametogenic failure, resulting in subfertility or infertility (*Jiao et al., 2021*; *Kanatsu-Shinohara and Shinohara, 2013*; *La and Hobbs, 2019*; *Song and*

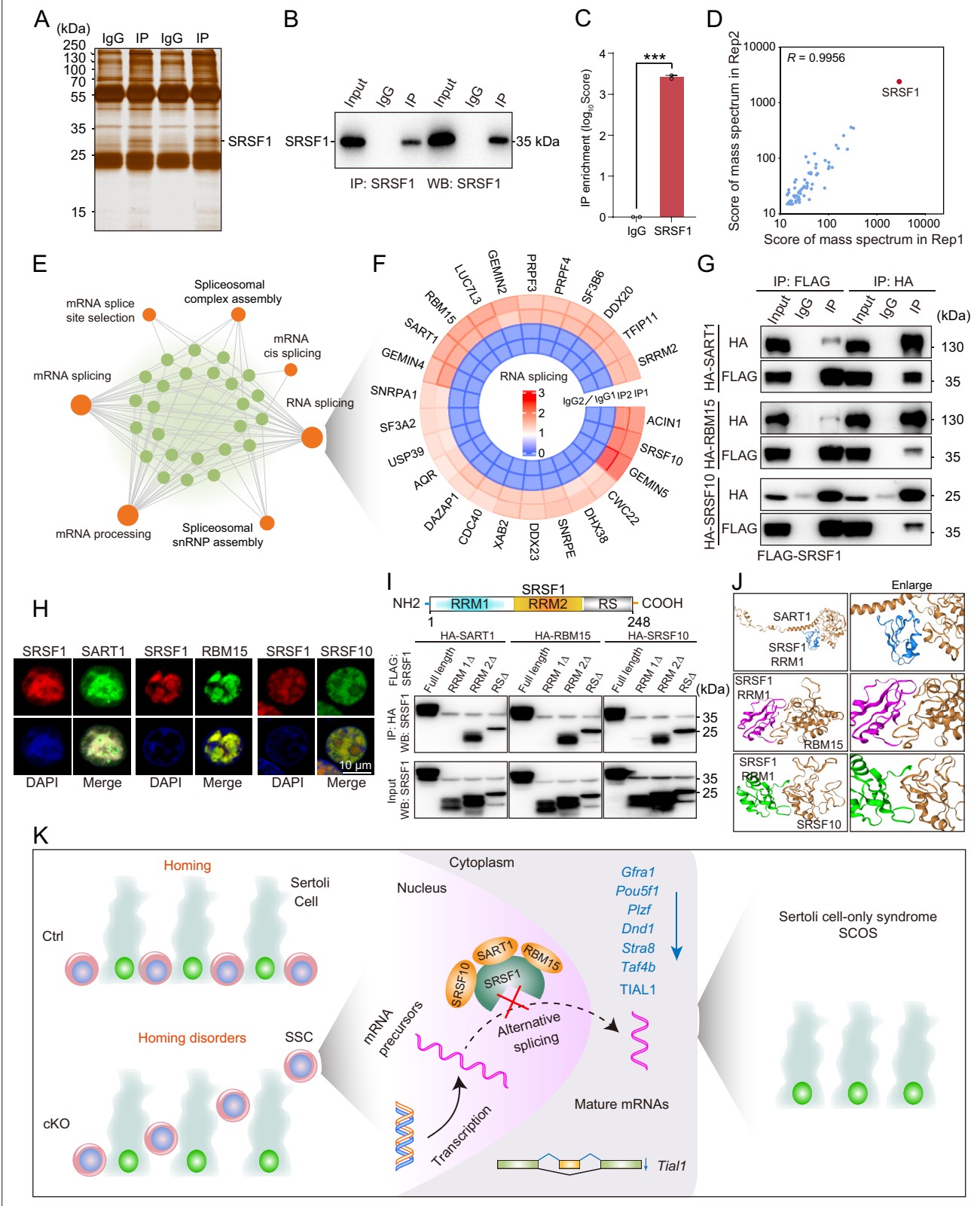

**Figure 9.** SRSF1 recruits alternative splicing (AS)-related proteins to modulate AS in testes. (**A**) Silver-stained gel of SRSF1 and control immunoprecipitates from 5 days postpartum (dpp) mouse testis extracts. (**B**) IP experiment was performed in 5 dpp mouse testis extracts. (**C**) IP of SRSF1 from immunoprecipitation mass spectrometry (IP-MS) data. (**D**) *Pearson's* correlation analysis showed the coefficient between the two samples for IP-MS data. (**E**) Network showing Gene Ontology (GO) enrichment analyses of SRSF1-binding proteins. (**F**) Circular heatmap of AS-related proteins.

*Figure 9 continued on next page*

*Figure 9 continued*

(**G**) Co-immunoprecipitation (Co-IP) experiment for detecting the SRSF1 association between SART1, RBM15, and SRSF10 in 293T cells. (**H**) SRSF1-mCherry cotransfected with SART1-eGFP, RBM15-eGFP, and SRSF10-eGFP 293T cells is shown. DNA was stained with DAPI. Scale bar, 10 µm. (**I**) Co-IP experiment for detecting the SRSF1-truncated protein association between SART1, RBM15, and SRSF10 in 293T cells. (**J**) A schematic diagram of protein interactions is shown. (**K**) Schematic illustration of the molecular mechanisms by which SRSF1 regulates homing of mouse precursor spermatogonial stem cells (SSCs).

The online version of this article includes the following source data for figure 9:

**Source data 1.** IP experiment was performed in 5 days postpartum (dpp) mouse testis extracts.

**Source data 2.** Co-immunoprecipitation (Co-IP) experiment for detecting the SRSF1 association between SART1, RBM15, and SRSF10 in 293T cells.

**Source data 3.** Co-immunoprecipitation (Co-IP) experiment for detecting the SRSF1-truncated protein association between SART1, RBM15, and SRSF10 in 293T cells.

*Wilkinson, 2014*). Loss-of-function mutations in humans and corresponding knockout/mutated mice have been extensively researched (*Jiao et al., 2021*). However, AS-related posttranscriptional regulation during meiosis has not been well studied. In recent years, there have been reports that the RNA-binding proteins SRSF10, UHRF1, BUD31, and BCAS2 regulate AS in mouse SSCs (*Liu et al., 2022*; *Liu et al., 2017*; *Qin et al., 2023*; *Zhou et al., 2022*). This study used a multiomics approach to perform in-depth analyses of SRSF1-mediated posttranscriptional regulatory mechanisms to enrich the field. It also provides new ideas and insights for clinical diagnosis and treatment.

In summary, this study demonstrates that SRSF1 plays a critical role in posttranscriptional regulation to implement homing of precursor SSCs (*Figure 9K*). Thus, the posttranscriptional regulation of SRSF1-mediated splicing is resolved during the formation of SSC pools and the establishment of niches.

## Materials and methods

### Mouse strains

C57BL/6N and ICR mice were purchased from Beijing Vital River Laboratory Animal Technology Co., Ltd. *Srsf1*[Fl/Fl] mice were generated in the laboratory of Prof. Xiangdong Fu (University of California, San Diego, CA, USA) and were kindly provided by Prof. Yuanchao Xue (Institute of Biophysics, Chinese Academy of Sciences, Beijing, China) (*Xu et al., 2005*). *Vasa*-Cre mice were obtained from The Jackson Laboratory (*Gallardo et al., 2007*). To generate *Srsf1* cKO mice, *Vasa*-Cre mice were crossed with *Srsf1*[Fl/Fl] mice. The primers used for PCR to genotype *Srsf1*[Fl/Fl] and *Vasa*-Cre mice are shown in *Supplementary file 5*. All mice were bred and housed under specific pathogen-free conditions with a controlled temperature (22 ± 1°C) and exposed to a constant 12 hr light-dark cycle in the animal facilities of China Agricultural University. All experiments were conducted according to the guidelines and with the approval of the Institutional Animal Care and Use Committee of China Agricultural University (No. AW80401202-3-3).

### Cell

HEK293T cells were obtained from the American Type Culture Collection (ATCC) and cultured in Dulbecco's modified Eagle's medium (C11995500BT, Gibco) supplemented with 10% fetal bovine serum (SE200-ES, VISTECH), Penicillin-Streptomycin Solution (C0222, Beyotime). All cells were maintained in a humidified incubator containing 5% $CO_2$ at 37°C. In addition, all of the cells used in the study tested negative in routine tests for *Mycoplasma* species using RT-PCR.

### Fertility test

For 15 days, two 8-week-old ICR female mice were caged with one 8-week-old male control (Ctrl) or cKO mouse. The mice were kept individually after the appearance of the vaginal plug, and the dates were recorded. Male mice continue to be caged after 2 days. The number of pups from each female was recorded each day, and the date of parturition was recorded.

## Immunostaining and histological analyses

Mouse testes were fixed with 4% paraformaldehyde (PFA, P6148-500G, Sigma-Aldrich) in PBS (pH 7.4) at 4°C overnight, dehydrated in graded ethanol solutions, vitrified with xylene, and embedded in paraffin. Testis sections were cut at a 5 μm thickness for immunostaining and histological analyses. For histological analyses, sections were dewaxed in xylene, rehydrated in a graded ethanol solution, and stained with haematoxylin-eosin. After sealing the slides with neutral resin, a Ventana DP200 system was used for imaging. For immunofluorescence analyses, antigen retrieval was performed by microwaving the sections with sodium citrate buffer (pH 6.0). After blocking with 10% normal goat serum at room temperature for 1 hr, the sections were incubated with primary antibodies (*Supplementary file 6*) in 5% normal goat serum overnight at 4°C. After washing with PBS, the sections were incubated with secondary antibodies (*Supplementary file 6*) at room temperature in the dark for 1 hr. The slides were mounted in an antifade mounting medium with DAPI (P0131, Beyotime). Photographs were taken with a Nikon A1 laser scanning confocal microscope and a Zeiss OPTOME fluorescence microscope. For immunohistochemistry analyses, antigen retrieval was performed by microwaving the sections with sodium citrate buffer (pH 6.0). Five dpp testis sections were prepared as described in the instructions for Immunohistochemistry Kit (PV-9001, ZSGB-BIO). After sealing the slides with neutral resin, a Ventana DP200 system was used for imaging.

## Whole-mount immunostaining

The testes were collected and dispersed with 5 ml syringes. Blown-out tubules were fixed in PFA at 4°C for 4 hr. The tubules were washed three times with PBS (pH 7.4) for 5 min each and stored at 4°C. The tubules were permeated with 0.3% Triton X-100 for 1 hr at 4°C. Then, whole-mount staining followed the immunofluorescence protocol.

## TUNEL apoptosis analyses

Seven dpp testis sections were prepared as described in the instructions for the TUNEL Apoptosis Assay Kit (C1088, Beyotime). Photographs were taken with a Nikon A1 laser scanning confocal microscope and a Zeiss OPTOME fluorescence microscope. Apoptosis rate was derived from the number of TUNEL-positive signals in tubules as a percentage of the total number of germ cells.

## RT-PCR and RT-qPCR

Total RNA was extracted by using RNAiso Plus (9109, Takara), and the concentration was measured with a Nano-300 ultramicro spectrophotometer (Allsheng). cDNA was obtained according to the instructions of a TIANScript II RT kit (KR107, TIANGEN). The expression of transcripts of the target gene was measured by using a LightCycle 96 instrument (Roche) with Hieff UNICON SYBR green master mix (11198ES08, Yeasen). AS analyses were performed on a RePure-A PCR instrument (BIO-GENER). Primers were synthesised by Sangon Biotech (*Supplementary file 5*). The expression level of *Gapdh* or *Actb* was used as the control, and this value was set as 1. Other samples' relative transcript expression levels were obtained by comparing them with the control results.

## RNA-seq

Total RNA was extracted from mouse testes according to the above protocol at 5 dpp. Briefly, mRNA was purified from total RNA using poly-T oligo-attached magnetic beads. After fragmentation, we established a transcriptome sequencing library and assessed library quality on an Agilent Bioanalyzer 2100 system. The clustering of the index-coded samples was performed on a cBot Cluster Generation System using a TruSeq PE Cluster kit v3-cBot-HS (Illumina) according to the manufacturer's instructions. After cluster generation, the library preparations were sequenced on the Illumina NovaSeq platform, and 150 bp paired-end reads were generated. After quality control, all downstream analyses were performed on clean, high-quality data. The reference genome index was built, and paired-end clean reads were aligned to the reference genome using HISAT2 software (version 2.0.5). FeatureCounts (version 1.5.0) counted the reads mapped to each gene. Then, the FPKM value of each gene was calculated based on the length of the gene and the read count mapped to this gene. Differential expression analyses of cKO/Ctrl mouse testes (three biological replicates per condition) were performed using the DESeq2 R package (version 1.20.0). Genes with a padj ≤0.05 identified by DESeq2 were considered differentially expressed.

## AS analyses

rMATS software (version 3.2.5) was used to analyse the AS events in cKO mouse germ cells based on RNA-seq data. Five types of AS events (SE, RI, MXE, A5SS, and A3SS) were revealed by rMATS software. Our threshold for screening differentially significant AS events was an FDR of less than 0.05. Splicing events with an FDR less than 0.05 and an inclusion-level difference with a significance of at least 5% (0.05) were considered statistically significant. IGV (2.16.0) software was used to visualise and confirm AS events based on RNA-seq data.

## GO enrichment analyses

The GO enrichment analyses of differentially expressed genes and AS genes were implemented with the clusterProfiler R package (version 3.4.4), in which gene length bias was corrected. All expressed genes (TPM sum of all samples ≥1) are listed background. GRCm38/mm10 was used as a mouse reference genome, and the Benjamini-Hochberg multiple methods was applied to adjust for multiple testing. GO enrichment analyses with corrected p values of less than 0.05 were significantly enriched for differentially expressed genes and AS genes.

## Western blotting

Total protein was extracted with cell lysis buffer (P0013, Beyotime) containing PMSF (1:100, ST506, Beyotime) and a protease inhibitor cocktail (1:100, P1005, Beyotime). Protein concentration was determined with the BCA Protein Assay Kit (CW0014S, CWBiotech). The protein lysates were electrophoretically separated on sodium dodecyl sulfate-polyacrylamide gels and electrically transferred to polyvinylidene fluoride membranes (IPVH00010, Millipore). The membranes were blocked in 5% skimmed milk for 1 hr and incubated with the primary antibodies (*Supplementary file 6*) for one night at 4°C. Then, the membranes were incubated with secondary antibodies (*Supplementary file 6*) at room temperature for 1 hr. The proteins were visualised using a Tanon 5200 chemiluminescence imaging system following incubation with BeyoECL Plus (P0018S, Beyotime).

## IP, IP-MS, Co-IP

Total protein was extracted with cell lysis buffer (P0013, Beyotime) containing PMSF (1:100, ST506, Beyotime) and a protease inhibitor cocktail (1:100, P1005, Beyotime). After incubation on ice for 20 min, the lysate was added to 20 µl of protein A agarose (P2051-2 ml, Beyotime) for pre-clearing at 4°C for 1 hr. Two micrograms of an SRSF1 antibody (sc-33652, Santa Cruz Biotechnology) and a normal mouse IgG (sc-3879, Santa Cruz Biotechnology) were added to the lysate and the mixture was incubated overnight at 4°C. The next day, 60 µl of protein A agarose was added to the lysate, which was then incubated at 4°C for 4 hr. The agarose complexes containing antibodies and target proteins were washed three times for 5 min at 4°C. IP and Co-IP were performed according to the above western blotting protocol. The complex was sent to the protein mass spectrometry laboratory for IP-MS analyses using a Thermo Q-Exactive high-resolution mass spectrometer (Thermo Scientific, Waltham, MA, USA). Raw data from the mass spectrometer were preprocessed with Mascot Distiller 2.4 for peak picking. The resulting peak lists were searched against the uniport mouse database using Mascot 2.5 search engine.

## RIP and RIP-qPCR

As described previously (*Gagliardi and Matarazzo, 2016*), RIP was performed using 5 dpp mouse testes. The testes were lysed in cell lysis buffer (P0013, Beyotime) containing PMSF (1:100, ST506, Beyotime), a proteinase inhibitor cocktail (1:100, P1005, Beyotime), DTT (1:50, ST041-2 ml, Beyotime), and an RNase inhibitor (1:20, R0102-10 kU, Beyotime). After incubation on ice for 20 min, the lysate was added to 20 µl of protein A agarose (P2051-2 ml, Beyotime) for pre-clearing at 4°C for 1 hr. Two micrograms of an SRSF1 antibody (sc-33652, Santa Cruz Biotechnology) and a normal mouse IgG (sc-3879, Santa Cruz Biotechnology) were added to the lysate, which was then incubated overnight at 4°C. The next day, 60 µl of protein A agarose was added to the lysate, and the mixture was incubated at 4°C for 4 hr. The agarose complexes containing antibodies, target proteins, and RNA were washed three times for 5 min at 4°C and repeated. Protein-bound RNA was extracted using RNAiso Plus and a Direct-zol RNA MicroPrep Kit. RIP-qPCR was performed according to the above RT-qPCR protocol.

## Statistical analyses

*Pearson's* correlation coefficients (R) were calculated by using the scores of the two samples for MS. GraphPad Prism software (version 9.0.0) was used for the statistical analyses, and the values and error bars represent the mean ± SEM. Significant differences between the two groups were analysed using Student's *t*-test. Statistical significance is indicated as follows: exact p value p ≥ 0.05; *p < 0.05; **p < 0.01; ***p < 0.001; ****p < 0.0001.

## Acknowledgements

This work was supported by the National Natural Science Foundation of China (32171111); and the Beijing Natural Science Foundation (5222015). We thank Prof. Yuanchao Xue (Institute of Biophysics, Chinese Academy of Sciences, Beijing, China) for sharing *Srsf1*^Fl/Fl mice, Prof. Shuyang Yu, Hua Zhang, and Chao Wang (China Agricultural University, Beijing, China) for thoughtful discussions and suggestions, and all the members of Prof. Hua Zhang, Chao Wang, and Shuyang Yu laboratory for helpful discussions and comments. We thank Novogene for their assistance with the RNA-seq and CLIP-seq experiments.

## Additional information

### Funding

| Funder | Grant reference number | Author |
|---|---|---|
| National Natural Science Foundation of China | 32171111 | Jiali Liu |
| Natural Science Foundation of Beijing Municipality | 5222015 | Jiali Liu |

The funders had no role in study design, data collection and interpretation, or the decision to submit the work for publication.

### Author contributions

Longjie Sun, Data curation, Software, Formal analysis, Methodology, Writing - original draft; Zheng Lv, Data curation, Methodology; Xuexue Chen, Rong Ye, Shuang Tian, Chaofan Wang, Xiaomei Xie, Lu Yan, Xiaohong Yao, Methodology; Yujing Shao, Sheng Cui, Writing – review and editing; Juan Chen, Data curation, Software, Writing – review and editing; Jiali Liu, Conceptualization, Data curation, Software, Funding acquisition, Writing – review and editing

### Author ORCIDs

Longjie Sun http://orcid.org/0000-0002-1285-3823
Sheng Cui http://orcid.org/0000-0002-3826-3768
Jiali Liu https://orcid.org/0000-0002-6214-569X

### Ethics

All experiments were conducted according to the guidelines and with the approval of the Institutional Animal Care and Use Committee of China Agricultural University (No. AW80401202-3-3).

Reviewer #1 (Public Review): https://doi.org/10.7554/eLife.89316.4.sa1
Reviewer #2 (Public Review): https://doi.org/10.7554/eLife.89316.4.sa2
Reviewer #3 (Public Review): https://doi.org/10.7554/eLife.89316.4.sa3
Author Response https://doi.org/10.7554/eLife.89316.4.sa4

## Additional files

### Supplementary files

• Supplementary file 1. SRSF1 peak-containing genes identified through crosslinking

immunoprecipitation and sequencing (CLIP-seq).

- Supplementary file 2. Differentially expressed genes were analysed in this study.
- Supplementary file 3. Alternative splicing (AS) events were analysed in conditional knockout (cKO) and Ctrl testes.
- Supplementary file 4. The SRSF1 interacting proteins were displayed through immunoprecipitation mass spectrometry (IP-MS).
- Supplementary file 5. Primer sequences were used in this study.
- Supplementary file 6. Antibodies were used in this study.
- Supplementary file 7. Mass spec data.
- MDAR checklist

### Data availability

All data generated or analysed during this study are included in this published article, its supplementary information files and publicly available repositories. Mass spec data are included in *Supplementary file 7* (Mass spec data). The RNA-seq data were deposited in GEO (https://www.ncbi.nlm.nih.gov/geo/) under accession number GSE227575.

The following dataset was generated:

| Author(s) | Year | Dataset title | Dataset URL | Database and Identifier |
|---|---|---|---|---|
| Sun LJ, Lv Z, Liu JL | 2024 | Splicing factor SRSF1 is essential for homing of precursor spermatogonial stem cells in mice | https://www.ncbi.nlm.nih.gov/geo/query/acc.cgi?acc=GSE227575 | NCBI Gene Expression Omnibus, GSE227575 |

The following previously published dataset was used:

| Author(s) | Year | Dataset title | Dataset URL | Database and Identifier |
|---|---|---|---|---|
| Sun LJ, Chen J, Ye R, Lv Z | 2024 | SRSF1 is crucial for male meiosis through alternative splicing during homologous pairing and synapsis in mice [CLIP-seq] | https://www.ncbi.nlm.nih.gov/geo/query/acc.cgi?acc=GSE227303 | NCBI Gene Expression Omnibus, GSE227303 |

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
