## [Editor Report · eLife assessment]

In this **valuable** study, the authors characterize the role of splicing factor SRSF1 during spermatogenesis with a conditional knockout of *Srsf1* in male germ cells. The phenotype and molecular role of SRSF1 in regulating alternative splicing in precursor spermatogonial stem cells in juvenile testes are **convincingly** supported. The paper also provides **convincing** evidence that the mRNA encoding Tial, a factor relevant to spermatogonial maintenance and male fertility, is alternatively spliced in testis and that this splicing is regulated by SRSF1. The work will be of interest to the fields of reproductive biology, stem cell biology, and alternative splicing.

---

## [Referee Report · Reviewer #1 (Public Review)]

In this study, the authors seek to characterize the role of splicing factor SRSF1 during spermatogenesis using Vasa-Cre;Srsf1Fl/del mice model. The authors first revealed that spermatogonia-related genes (e.g., Plzf, Id4, Setdb1, Stra8, Tial1/Tiar, Bcas2, Ddx5, Srsf10, Uhrf1, and Bud31) were bound by SRSF1 in the mouse testes by CLIP-seq. The authors convincingly demonstrated that specific deletion of SRSF1 in mouse gem cells with vasa-cre lead to NOA by impairing homing and failure survival of spermatogonia. To investigate the molecular mechanisms of SRSF1 in spermatogonia, further multiomics analysis including CLIP-seq, IP-MS, and RNA-seq were conducted. The results showed that SRSF1 coordinated with other RNA splicing-related proteins to directly bind and regulate the expression of nine spermatogonia-related genes especially Tial1/Tiar via alternative splicing. The authors revealed the critical role of SRSF1-mediated AS in precursor SSCs homing and survival, which may provide a framework to elucidate the molecular mechanisms of the posttranscriptional network underlying the formation of SSC pools and the establishment of niches. This work will be of interest to stem cell and reproductive biologists. The experiments are well-designed and conducted, and the overall methods and results are convincing except for the claim that altered splicing of the Tial1 transcript mediates the effect of SRSF1 loss.

---

## [Referee Report · Reviewer #2 (Public Review)]

Summary

The authors seek to characterize the role of splicing factor SRSF1 during spermatogenesis. Using a conditional deletion of Srsf1 in germ cells, they find that SRSF1 is required for male fertility. Via immunostaining and RNA-seq analysis of the Srsf1 conditional knockout (cKO) testes, combined with SRSF1 CLIP-seq and IP-MS data from the testis, they ultimately conclude that Srsf1 is required for homing of precursor spermatogonial stem cells (SCCs) due to alternative splicing.

Strengths

The overall methods and results are robust. The histological analysis of the Srsf1 cKO traces the origins of the fertility defect to the postnatal testis, and the authors have generated interesting datasets characterizing SRSF1's RNA targets and interacting proteins specifically in the testis.

Ultimately, the authors have shown that SRSF1's effects on alternative splicing are required to establish spermatogenesis. In the absence of Srsf1, the postnatal gonocytes do not properly mature into spermatogonia and consequently never initiate spermatogenesis.

---

## [Referee Report · Reviewer #3 (Public Review)]

In this study, Sun et al examine the role of the splicing factor SRSF1 in spermatogenesis in mice. Alternative splicing is important for spermatogenic development, but its regulation and major developmental roles during spermatogenesis are not well understood. The authors set out to better define both SRSF1 function in testes and the contribution of alternative splicing. They generate several large 'omics datasets to define SRSF1 targets in testis, including RNA interactions by CLIP-seq in whole testis, protein interactions by IP-mass spec in whole testis, and RNA sequencing to detect expression levels and splice variants. They also examine the phenotype of germline conditional knockouts (cKO) for Srsf1, using the early-acting Vasa-Cre, and find a severe depletion of germ cells starting at 7 days post partum (dpp) and culminating with a lack of germ cells (Sertoli Cell Only Syndrome) by adulthood. They detect differences in gene expression as well as differences in splicing between control and knockout, including 9 genes that are downregulated, experience alternative splicing, and whose transcripts are also bound by SRSF1, and identify the Tial1/Tiar transcript as one of these targets. They conclude that SRSF1 is required for homing and self-renewal of precursor spermatogonial stem cells, and suggest that this role may be mediated in part though its regulation of Tial1/Tiar splicing.

Strengths of the paper include detailed phenotyping of the Srsf1 cKO, which convincingly supports the Sertoli Cell Only phenotype, establishes the timing of the first appearance of the spermatogonial defect, and provides new insight into the role of splicing factors and SRSF1 specifically in spermatogenesis. Another strength is the generation of CLIP-seq, IP-MS, and RNA-seq datasets which will be a useful resource for the field of germ cell development. Overall, the results support the claims made. While the study does not provide a full mechanistic understanding of how alternative splicing mediated by SRSF1 affects SSC precursors, the contributions are novel and useful, and will be of interest to the fields of alternative splicing and male reproductive biology.

---

## [Author Response]

The following is the authors’ response to the previous reviews.

**Reviewer #3 (Recommendations For The Authors):**
1. Fig. 2B: In their previous comment #6, I assume that Reviewer #2 was asking about peaks that were called as statistically significant above background, not just "higher" as assessed by eye. The authors have now marked peaks that are "higher" but still do not indicate that they were called as statistically significant by any software. I agree that they need to indicate in the figure which peaks were discovered by formal analysis.

Response: Thank you for the professional suggestions. We used the Piranha (version 1.2.1) software to call peaks from CLIP-seq data, in which the P-value threshold for peaks (i.e., the -p parameter) was set as 0.05. And then any region above the IgG peak could be a binding region, and of course, the higher the peak, the more pre-mRNA SRSF1 binds in that region.

1. Similar to the above comment, in Fig. 7G "visual analysis" of IGV tracks is not an assay. It is fine to show the tracks as an example of the differential expression called using DESeq2, but this should be described for what it is.

Response: We thank the reviewer for the professional comments. Following this advice, we have corrected the text in this revised version (Page 11, Line 233).

1. Fig 5C: TUNEL results are supported by a single image of only a few cells. It is important to include quantitation as has been done for other microscopy data.

Response: Thank you for the professional suggestions. Following this advice, we have added the quantitative data in Figure 5C. Also, we have added specific quantification methods to the text (Page 23, Line 484-485).

1. Legend to Fig 6C-E: I assume n=4 refers to the number of animals. It would be best to also know many cells/tubules were counted for each animal.

Response: Thank you for the helpful comments. Following this advice, we have revised the legend for Figure 6D, E (Page 12, Line 246-249).

1. There appears to be a mistake in line 285-287, which reads: "the overall analysis of aberrant AS events showed that SRSF1 effectively promotes the occurrence of SE and MXE events and inhibits the occurrence of RI events." The data in Fig 8C appears to show the opposite, with more SE and MXE, and fewer RI events, in the SRSF1 KO. This would imply that SRSF1 normally inhibits SE/MXE and promotes RI.

Response: Thank you very much for the professional comments. Following this advice, we have corrected the text in this revised version (Page 14, Line 286-288).

1. In Fig. 8E, an upper band is depleted in SRSF1 KO, but in Figure 8J, a much lower band is depleted. How is this explained?

Response: Thank you for the professional suggestions. Since exon 7 of Tial1 is in the non-coding region, the lower band in Figure 8E does not correspond to the lower band in Figure 8J. For better understanding, we show the detailed information of Tial1 in the attached Figure S3.

1. Line 81: As a very minor point, "AS" is defined as alternative splicing in the abstract, but should be re-defined again in the main text when first mentioned.

Response: Thank you for the helpful comments. Following this advice, we have corrected the text in this revised version (Page 3, Line 81).